# The Effect of Furanocoumarin Derivatives on Induction of Apoptosis and Multidrug Resistance in Human Leukemic Cells

**DOI:** 10.3390/molecules24091824

**Published:** 2019-05-12

**Authors:** Tomasz Kubrak, Marcin Czop, Przemysław Kołodziej, Marta Ziaja-Sołtys, Jacek Bogucki, Anna Makuch-Kocka, David Aebisher, Janusz Kocki, Anna Bogucka-Kocka

**Affiliations:** 1Department of Biochemistry and General Chemistry, Faculty of Medicine, University of Rzeszow, Aleja Rejtana 16A, 35-310 Rzeszów, Poland; kubrak.tomasz@gmail.com; 2Department of Clinical Genetics, Faculty of Medicine with Dentistry Division, Medical University of Lublin, 20-059 Lublin, Poland; marcin.czop@umlub.pl (M.C.); jacekbogucki@wp.pl (J.B.); januszkocki@umlub.pl (J.K.); 3Chair and Department of Biology and Genetics, Faculty of Pharmacy with Medical Analytics Division, Medical University of Lublin, 20-059 Lublin, Poland; przemyslaw.kolodziej@umlub.pl (P.K.); marta.ziaja-soltys@umlub.pl (M.Z.-S.); 4Department of Pharmacology, Faculty of Health Sciences, Medical University of Lublin, 20-059 Lublin, Poland; anna.makuch@poczta.fm; 5Department of Photomedicine and Physical Chemistry, Faculty of Medicine, University of Rzeszow, Aleja Rejtana 16A, 35-310 Rzeszów, Poland; daebisher@ur.edu.pl

**Keywords:** imaging flow cytometry, apoptosis, leukemic cell lines, furanocoumarin, multidrug resistance

## Abstract

Background: The insensitivity of cancer cells to therapeutic agents is considered to be the main cause of failure of therapy and mortality of patients with cancer. A particularly important problem in these patients is the phenomenon of multidrug resistance, consisting of abnormal, elevated expression of transport proteins (ABC family). The aim of this research included determination of IC50 values of selected furanocoumarins in the presence and absence of mitoxantrone in leukemia cells and analysis of changes in apoptosis using anexinV/IP and Casp3/IP after 24 h exposure of cell lines to selected coumarins in the presence and absence of mitoxantrone in IC50 concentrations. Methods: Research was conducted on 3 cell lines derived from the human hematopoietic system: HL-60, HL-60/MX1 and HL-60/MX2. After exposure to coumarin compounds, cells were subjected to cytometric analysis to determine the induction of apoptosis by two methods: the Annexin V test with propidium iodide and the PhiPhiLux-G1D2 reagent containing caspase 3 antibodies. Results: All of the furanocoumarin derivatives studied were found to induce apoptosis in leukemia cell lines. Conclusions: Our results clearly show that the furanocoumarin derivatives are therapeutic substances with antitumor activity inducing apoptosis in human leukemia cells with phenotypes of resistance.

## 1. Introduction

Coumarins are a group of biologically active compounds that are produced by living organisms (plants, fungi and bacteria) as secondary metabolites. They are of great interest among researchers as coumarin agents possess numerous pharmacological properties including anti-inflammatory, antithrombotic, antibacterial, antifungal, antiviral, anti-hypertensive, anticonvulsant and anticancer, among others [1,2,3,4,5].

The four main classes of coumarins are simple coumarins, furanocoumarins, pyrano coumarins and pyrone substituted coumarins [6]. Furanocoumarins are a specific group of secondary metabolites that are commonly found in higher plants. Biochemical modifications during furanocoumarin-coumarin biosynthesis include prenylation reactions catalyzed by cytochrome P450 enzymes with subsequent formation of furan rings [7].

Furanocoumarins, in particular, have a strong antiproliferative effect, inhibiting the growth of cancer cells by modifying several molecular pathways, such as regulation of the signal transducer and activator of transcription 3, nuclear factor-kB, phosphatidylinositol-3-kinase/AKT and mitogen-activated protein kinase expression [7]. 

The process of programmed cell death or apoptosis is generally characterized by the distinctness of morphological traits. Apoptosis is considered an important component of various processes, including normal cell turnover, proper development and functioning of the immune system, embryonic development and chemically induced cell death. Uncontrolled apoptosis is responsible for numerous physiological changes in organisms, including the occurrence of neurodegenerative diseases, ischemic defects, autoimmune disorders and many types of cancers [8,9,10].

Since the introduction of methods based on flow cytometry, the number of studies on apoptosis has increased significantly. In this technique, propidium iodide (PI) dye is widely used in combination with Annexin V to determine whether cells are viable, apoptotic or necrotic through differences in cell membrane integrity and permeability [11]. The use of annexin V/PI dye system is a widely used approach to the study of apoptotic cells [12,13]. 

The ability of PI to penetrate the cell depends on the permeability of the membrane as PI does not stain live or early apoptotic cells due to the presence of an intact plasma membrane [11,14]. During late apoptosis and necrosis, the integrity of nuclear membranes decrease [15], allowing PI to pass through membranes, intercalate to nucleic acids and display red fluorescence [11,16].

Caspase 3 is considered to be the most important of the executioner caspases and is activated by any initiator caspases (caspase 8, caspase 9, or caspase 10). Caspase 3 specifically activates the endonuclease CAD. In proliferating cells, CAD is complexed with its inhibitor, ICAD. In apoptotic cells, activated caspase 3 cleaves ICAD to release CAD. CAD then degrades chromosomal DNA within nuclei and causes chromatin condensation. Caspase 3 also induces cytoskeletal reorganization and disintegration of the cell into apoptotic bodies [8].

Knowledge of the process of apoptosis offers the possibility of modulating cell life or death, which has significant therapeutic potential. Numerous studies focus on the analysis of the cell cycle by explaining signaling pathways that directly control the action of the cell cycle and its arrest. 

## 2. Results

### 2.1. Analysis of Cytotoxicity

The cytotoxicity of the coumarins tested was assessed by means of tryptan-blue staining in the presence of mitoxantrone (cytostatic (+C)) and absence of mitoxantrone. The experiment was carried out in triplicate and the average values were calculated from the given values. Diverse dose-dependent cytotoxicity of the IC50 (and IC50 (+C) cells of the three cancer cell lines exposed to coumarin derivatives were the subject of our previous work [17].

The addition of mitoxantrone is in accordance with the American Type Culture Collection (ATCC) manufacturer’s protocol and does not affect cell viability. 

### 2.2. Analysis of Apoptosis Induced by Coumarin Compounds at Doses of IC50 in the Presence of Mitoxantrone (+C) and without Mitoxantrone Using AnnexinV/IP

Based on the positive reaction of the cell membrane with Annexin V (A) and the intercalation of propidium iodide (IP) with DNA, four cell populations were distinguished: live A (–), IP (–), early apoptotic A (+), IP (–), late apoptotic A (+), IP (+) and necrotic A (–), IP (+) (Figure 1 and Figure 2).

Statistically significant differences between the individual lines exposed to the tested furanocoumarin compounds without the presence of mitoxantrone (Table 1). Seven were found in a population of living cells; there were eight in the early apoptotic cell population, there were six in the late apoptotic cell population and seven in the populations of necrotic cells found.

There were statistically significant differences (Figure 3A, Table 1) in the number of live cells between the HL60 and HL60/MX1 lines after exposure to xanthotoxin (decreasing number of viable cells in HL60/MX1), heraclenin and byakangelicol (increasing live cells in HL60/MX1).

There were statistically significant differences in the number of live cells between the HL60 and HL60/MX2 only after exposure to bergapten (decreasing amount of viable cells in HL60/MX2).

There were statistically significant differences in the number of live cells between HL60/MX1 and HL60/MX2 after exposure to xanthotoxol, isopimpinellin and byakangelicin (decreasing amount of viable cells in HL60/MX2).

There were statistically significant differences (Figure 3C, Table 1) in the number of early apoptotic cells between the HL60 and HL60/MX1 lines after exposure to only byakangelicol (increasing number of early apoptotic cells in HL60/MX1).

There were statistically significant differences in the number of early apoptotic cells between the HL60 and HL60/MX2 lines after exposure to isopimpinellin, bergapten, byakangelicin (increasing number of early apoptotic cells in HL60/MX2) and phellopterin (decreasing number of early apoptotic cells in HL60/MX2).

There were statistically significant differences in the number of early apoptotic cells between the HL60/MX1 and HL60/MX2 lines after exposure to xanthotoxol, xanthotoxin (increasing number of early apoptotic cells in HL60/MX2) and heraclenin (decrasing number of early apoptotic cells in HL60/MX2).

There were statistically significant differences (Figure 3E, Table 1) in the number of late apoptotic cells between the HL60 and HL60/MX1 lines after exposure to xanthotoxin (increasing number of late apoptotic cells in HL60/MX1) and heraclenin (decreasing number of late apoptotic cells in HL60/MX1).

There were no statistically significant differences in the number of late apoptotic cells between the HL60 and HL60/MX2 lines.

There were statistically significant differences in the number of late apoptotic cells between the HL60/MX1 and HL60/MX2 lines after exposure to xanthotoxol, isopimpinellin, bergapten and byakangelicin (increasing number of late apoptotic cells in HL60/MX2).

There were statistically significant differences (Figure 3G, Table 1) in the number of necrotic cells between the HL60 and HL60/MX1 lines after only exposure to heraclenin and isopimpinellin (decreasing number of necrotic cells in HL60/MX1).

There were no statistically significant differences in the number of necrotic cells between the HL60 and HL60/MX2 lines.

There were statistically significant differences in the number of necrotic cells between the HL60/MX1 and HL60/MX2 lines after exposure to xanthotoxol, bergapten, byakangelicin, byakangelicol and phellopterin (increasing number of necrotic cells in HL60/MX2).

Statistically significant differences between the individual lines exposed to the tested furanocoumarin compounds with the presence of mitoxantrone (+C) (Table 2). Seven were found in a population of living cells; there were four in the early apoptotic cell population, there were eight in the late apoptotic cell population and five in the populations of necrotic cells found.

There were statistically significant differences (Figure 3B, Table 2) in the number of live cells between the HL60 and HL60/MX1 lines after exposure to heraclenin+C (increasing number of live cells in HL60/MX1) and isopimpinellin+C (decreasing number of live cells in HL60/MX1).

There were statistically significant differences in the number of living cells between the HL60 and HL60/MX2 lines after exposure to xanthotoxin+C, bergapten+C and phellopterin+C (increasing number of live cells in HL60/MX2) and byakangelicol+C (decreasing number of live cells in HL60/MX2).

There were statistically significant differences in the number of live cells between the HL60/MX1 and HL60/MX2 lines after exposure to only the byakangelicin+C (increasing number of live cells in HL60/MX2).

There were statistically significant differences (Figure 3D, Table 2) in the number of early apoptotic cells between the HL60 and HL60/MX1 lines after exposure to only heraclenin+C (increasing number of early apoptotic cells in HL60/MX1).

There were statistically significant differences in the number of early apoptotic cells between the HL60 and HL60/MX2 lines after exposure to only byakangelicin+C (increasing number of early apoptotic cells in HL60/MX2).

There were statistically significant differences in the number of early apoptotic cells between the HL60/MX1 and HL60/MX2 lines after exposure to xanthotoxol+C and bergapten+C (increasing number of early apoptotic cells in HL60/MX2).

There were statistically significant differences (Figure 3F, Table 2) in the number of late apoptotic cells between the HL60 and HL60/MX1 lines after exposure to heraclenin+C (decreasing number of late apoptotic cells in HL60/MX1) and isopimpinellin+C (increasing number of late apoptotic cells in HL60/MX1).

There were statistically significant differences in the number of late apoptotic cells between the HL60 and HL60/MX2 lines after exposure to xanthotoxol+C, xanthotoxin+C and phellopterin+C (decreasing number of late apoptotic cells in HL60/MX2) and byakangelicol+C (increasing number of late apoptotic cells in HL60/MX2).

There were statistically significant differences in the number of late apoptotic cells between the HL60/MX1 and HL60/MX2 lines after exposure to bergapten+C and byakangelicin+C (decreasing number of late apoptotic cells in HL60/MX2).

There were statistically significant differences (Figure 3H, Table 2) in the number of necrotic cells between the HL60 and HL60/MX1 lines only after exposure to xanthotoxol+C (decreasing number of necrotic cells in HL60/MX1).

There were statistically significant differences in the number of necrotic cells between the HL60 and HL60/MX2 lines after exposure to xanthotoxin+C (increasing number of necrotic cells in HL60/MX2), bergapten+C and byakangelicol+C (decreasing number of necrotic cells in HL60/MX2).

There were statistically significant differences in the number of necrotic cells between the HL60/MX1 and HL60/MX2 lines only after exposure to byakangelicin+C (decreasing number of necrotic cells in HL60/MX2).

### 2.3. Analysis of Apoptosis Induced by Coumarin Compounds in an IC50 Dose in the Presence of Mitoxantrone (+C) and without Mitoxantrone Using Caspase 3

Using the flow cytometry method and an antibody recognizing the active form of caspase 3 and propidium iodide, an analysis of induction of apoptosis by coumarin compounds at IC50 and IC50+C doses was performed.

Based on the reaction of cells with the antibody against caspase 3 and propidium iodide, four cell fractions were distinguished: live Casp3(–)/PI(–), early apoptotic Casp3(+)/PI(–), late apoptotic Casp3(+)/IP(+) and necrotic Casp3(–)/IP(+) (Figure 4 and Figure 5).

Statistically significant differences between the individual lines exposed to the tested furanocoumarin compounds without the presence of mitoxantrone (Table 3). Eight were found in a population of living cells; there were five in the early apoptotic cell population, there were eight in the late apoptotic cell population and eight in the populations of necrotic cells found.

There were statistically significant differences (Figure 6A, Table 3) in the number of life cells between the HL60 and HL60/MX1 lines after exposure to xanthotoxol, heraclenin, byakangelicin and byakangelicol (increasing number of viable cells in HL60/MX1).

There were statistically significant differences in the number of live cells between the HL60 and HL60/MX2 after exposure to xanthotoxin and phellopterin (increasing amount of viable cells in HL60/MX2).

There were statistically significant differences in the number of life cells between the HL60/MX1 and HL60/MX2 after exposure to isopimpinellin and bergapten (decreasing amount of viable cells in HL60/MX2).

There were no statistically significant differences (Figure 6C, Table 3) in the number of early apoptotic cells between the HL60 and HL60/MX1 lines.

There were statistically significant differences in the number of early apoptotic cells between the HL60 and HL60/MX2 lines after exposure to xanthotoxol, xanthotoxin and heraclenin (increasing number of early apoptotic cells in HL60/MX2).

There were statistically significant differences in the number of early apoptotic cells between the HL60/MX1 and HL60/MX2 lines after exposure to bergapten (increasing number of early apoptotic cells in HL60/MX2) and phellopterin (decrasing number of early apoptotic cells in HL60/MX2).

There were statistically significant differences (Figure 6E, Table 3) in the number of late apoptotic cells between the HL60 and HL60/MX1 lines after exposure to xanthotoxol, heraclenin and byakangelicin (decreasing number of late apoptotic cells in HL60/MX1).

There were statistically significant differences in the number of late apoptotic cells between the HL60 and HL60/MX2 lines after exposure to xanthotoxin, byakangelicol and phellopterin (decreasing number of late apoptotic cells in HL60/MX2).

There were statistically significant differences in the number of late apoptotic cells between the HL60/MX1 and HL60/MX2 lines after exposure to isopimpinellin and bergapten (increasing number of late apoptotic cells in HL60/MX2).

There were no statistically significant differences (Figure 6G, Table 3) in the number of necrotic cells between the HL60 and HL60/MX1 lines.

There were statistically significant differences in the number of necrotic cells between the HL60 and HL60/MX2 lines after only exposure to xanthotoxol, xanthotoxin, isopimpinellin, byakangelicol and phellopterin (increasing number of necrotic cells in HL60/MX2).

There were statistically significant differences in the number of necrotic cells between the HL60/MX1 and HL60/MX2 lines after exposure to heraclenin, bergapten and byakangelicin (increasing number of necrotic cells in HL60/MX2).

Statistically significant differences between the individual lines exposed to the tested furanocoumarin compounds with the presence of mitoxantrone (+C) (Table 4). Seven were found in a population of living cells; there were six in the early apoptotic cell population, there were eight in the late apoptotic cell population and eight in the populations of necrotic cells found.

There were statistically significant differences (Figure 6B, Table 4) in the number of live cells between the HL60 and HL60/MX1 lines after exposure to xanthotoxol+C, heraclenin+C, bergapten+C and byakangelicin+C (increasing number of live cells in HL60/MX1).

There were statistically significant differences in the number of living cells between the HL60 and HL60/MX2 lines after exposure to byakangelicol+C (increasing number of live cells in HL60/MX2) and phellopterin+C (increasing number of live cells in HL60/MX2).

There were statistically significant differences in the number of live cells between the HL60/MX1 and HL60/MX2 lines after exposure to only the isopimpinellin+C (decreasing number of live cells in HL60/MX2).

There were statistically significant differences (Figure 6D, Table 4) in the number of early apoptotic cells between the HL60 and HL60/MX1 lines after exposure to only phellopterin+C (increasing number of early apoptotic cells in HL60/MX1).

There were statistically significant differences in the number of early apoptotic cells between the HL60 and HL60/MX2 lines after exposure to xanthotoxol+C, xanthotoxin+C, heraclenin+C and byakangelicol+C (increasing number of early apoptotic cells in HL60/MX2).

There were statistically significant differences in the number of early apoptotic cells between the HL60/MX1 and HL60/MX2 lines after exposure to only isopimpinellin+C (increasing number of early apoptotic cells in HL60/MX2).

There were statistically significant differences (Figure 6F, Table 4) in the number of late apoptotic cells between the HL60 and HL60/MX1 lines after exposure to xanthotoxol+C, heraclenin+C, isopimpinellin+C, bergapten+C and byakangelicin+C (decreasing number of late apoptotic cells in HL60/MX1).

There were statistically significant differences in the number of late apoptotic cells between the HL60 and HL60/MX2 lines after exposure to xanthotoxin+C, byakangelicol+C and phellopterin+C (decreasing number of late apoptotic cells in HL60/MX2).

There were no statistically significant differences in the number of late apoptotic cells between the HL60/MX1 and HL60/MX2 lines.

There were no statistically significant differences (Figure 6H, Table 4) in the number of necrotic cells between the HL60 and HL60/MX1 lines.

There were statistically significant differences in the number of necrotic cells between the HL60 and HL60/MX2 lines after exposure to isopimpinellin+C, bergapten+C, byakangelicin+C, byakangelicol+C and phellopterin+C (increasing number of necrotic cells in HL60/MX2).

There were statistically significant differences in the number of necrotic cells between the HL60/MX1 and HL60/MX2 lines only after exposure to xanthotoxol+C, xanthotoxin+C and heraclenin+C (increasing number of necrotic cells in HL60/MX2).

## 3. Discussion

Resistance of cancer cells to chemotherapy is one of the causes of ineffectiveness of current pharmacotherapy. Modern medicine develops numerous strategies that would allow to overcome the resistance of cancer cells to the therapies used, as well as to learn the mechanisms that lead to the phenomenon of such resistance. The main problem is that cancer cells do not develop resistance based on one mechanism, but include a combination of several ways to escape from therapeutic strategies. Therefore, new substances, often of floral origin, are being sought, which effectively act as an inhibitor of proteins that affect MDR tumors. [18,19].

The cell lines tested in our previous work show high sensitivity to furanocoumarin compounds at doses of IC50 without mitoxantrone and IC50 with mitoxantrone (+C). The studied furanocoumarins were characterized by lower cytotoxicity than coumarins [17]. Analogous results were obtained by Yang et al. [20], who in their studies also showed a significantly higher cytotoxicity of simple coumarins than that of furanocoumarins. Numerous studies show that plant compounds, including coumarins, are often checked for cytotoxicity towards cells, and different tumor lines [21,22,23,24].

Thanks to the ability to analyze the image of cells in real time (FlowSight cytometer), characteristic changes in the examined cells were observed, consisting in the reduction of size and creation of apoptotic bodies. These characteristics and the positive results of the test with Annexin V/IP and caspase 3 indicate the induction of apoptosis in the cell by the test compounds.

Interesting results were also obtained during previous studies Kubrak et. al. [17]. Based on the results of the analysis of MDR, BCRP, LRP and MRP gene expression in leukemia cells exposed to the compounds tested, furanocoumarin compounds have been found to have a more promising mechanism of action.

Colon cancer cells (line SW-480) were exposed to three coumarins: 5-geranyloxy-7-methoxycoumarin, limetin and isopimpinelin obtained from *Citrus aurantifolia* extract (Christm.). All compounds inhibited the proliferation of SW-480 cells. The highest efficiency was reported for 5-geranyloxy-7-methoxycoumarin, the lowest for isopimpinellin. The inhibition of cell proliferation was associated with the induction of apoptosis, as evidenced by the results of the Annexin V assay and DNA fragmentation. Coumarin derivatives caused cell cycle arrest in the G0/G1 phase and induced apoptosis by activating the suppressor p53 gene, caspase 8 and 3, regulation of Bcl2 and inhibition of p38 phosphorylation [25].

Panno et al. [26] exposed MCF-7 breast cancer cells (human breast adenocarcinoma cell line) and SKBR-3 (cancer breast cancer line) to bergapten. Bergapten, regardless of photoactivation, stopped the cell cycle in the G0/G1 phase, introducing breast cancer cells into the apoptosis path and counteracting the stimulating effect of IGF-I/E2 on the growth of MCF-7 cells. Other team studies, conducted on human MCF-7 breast cancer cells, ZR-75 and SKBR-3, confirmed the anti-proliferative effect and induction of apoptosis by bergapten and UV-activated bergaptin [27].

Recent team research shows the inducing effect of bergaptene on metabolic reprogramming of MCF-7 and ZR75 breast cancer cells. Bergapten blocks glycolysis and significantly lowers glucose-6-phosphate dehydrogenase. Therapy with bergaptene causes changes in the metabolic pathways inducing cell death [28].

Yang et al. [20] studied the effect of osteol, emperorin, bergapten, isopimpinine and xanthoxin on cells: leukemias (HL-60 lineage), cervical cancer (HeLa line), colon cancer (CoLo 205 line) and normal PBMCs (peripheral blood mononuclear cells). They noticed that the highest cytotoxic activity is manifested by ostol and this is related to the construction, in this case with the presence of the prenyl group. Imperatorin showed the highest sensitivity to HL-60 line cells and the lowest toxicity to normal cells. Ostol and imperatorin cause the formation of apoptotic bodies and DNA fragmentation as well as increased PARP degradation in HL-60 cells [20].

The induction of apoptosis and cell cycle arrest was observed during the action of xantoxylin on gastric cancer cells line SGC-7901. It is noted that this action is associated with DNA damage. Apoptosis was caused by damage to the mitochondria, and the cell cycle is stopped in the S phase [29]. Studies were carried out with the use of xantotoxin, which stimulated the cells of the Jurkat leukemia line and normal lymphocytes. The use of this furanocoumarin caused an increase in the expression of caspase 8, 9, 3 and 7, which confirms apoptotic cell death [30].

Research by Yu-Ying Zhang et al. [31] clearly indicates the pro-apoptotic effect of coumarin compounds on MG63 cells (Human osteosarcoma). Exposure of MG63 cells to the coumarin compound caused a decrease in anti-apoptotic Bcl-2 protein, an increase in proapoptotic Bax protein and activation of caspase 3, 8 and 9. The obtained results confirm the antitumor properties of coumarins and cell death by apoptosis [31].

The high activity of coumarin compounds seems to be the basis for the design of new analogues characterized by pharmacokinetic changes, and thus increased activity and safety of use. The introduction of various substituents on the ring influences biological activity [32,33]. The challenge is for scientists is to create new drugs based on the design and synthesis of new derivatives with high activity and to determine their mechanism of action. Current progress in the design of new compound structures may lead to the discovery of a new anti-cancer drug [34].

Increased cancer mortality and high treatment costs are an impulse for the constant search for anticancer drugs with increased effectiveness.

## 4. Material and Methods

### 4.1. Cell Lines and Cell Culture

Human acute promyelocytic leukemia cell lines: HL60, HL60/MX1, HL60/MX2 were used. Cell lines were obtained from American Type Culture Collection (ATCC) 10801, University Boulevard Manassas, VA 20110, USA.

HL-60 (CCL 240)—is a promyelocytic cell line derived by S.J. Collins et al. The peripheral blood leukocytes were obtained by leukopheresis from a 36-year-old Caucasian female with acute promyelocytic leukemia.

HL-60/MX1 (CRL–2258)—a mitoxantrone resistant derivative of the HL-60 cell line was obtained from peripheral blood leukocytes obtained by leukopheresis from a patient with acute promyelocytic leukemia.

HL-60/MX2 (CRL–2257)—is also a mitoxantrone resistant derivative of the HL-60 cell line. HL-60/MX2 cells display atypical multidrug resistance (MDR) with absence of P-gp overexpression and altered topoisomerase II catalytic activity and reduced levels of topoisomerase II alpha and beta proteins.

The cells were maintained in RPMI 1640 medium (PAA Laboratories, Linz Austria) supplemented with 10% fetal bovine serum (FBS) (PAA Laboratories) for HL60/MX1, HL60/MX2 cell lines and 20% FBS for HL60 cells, penicillin-streptomycin (100U/mL PAA Laboratories) and amphotericin (PAA Laboratories) at 37 °C in a humidified atmosphere of 5% CO_2_.

### 4.2. Analysis of Cell Viability

Cells were seeded on 12-well plates (Sarstedt, Wr. Neudorf, Austria) at an initial density of 1 × 10^6^ cells/mL. After 24 h, the cell suspension was stimulated with coumarin-derivatives at concentrations ranging from 10 µM to 1000 µM. After 24 h, 1 mL of cell suspension was centrifuged at 1000 rpm for 5 min and the supernatant was discarded. The cells were resuspended in 50 µL PBS. From each tube, a 10 µL-cell suspension was taken and mixed with 10 µL of Trypan blue reagent (Bio-Rad, Hercules, CA, USA). The sample was incubated for 5 min. Cell viability was measured with a TC20 Automated Cell Counter (Bio-Rad, Hercules, CA, USA). Each experiment was repeated three times.

### 4.3. Chemicals

Xanthotoxol (Xan-l), xanthotoxin (Xan-n), heraclenin (Her), isopimpinellin (Iso), bergapten (Ber), byakangelicin (Bya-n), byakangelicol (Bya-l), phellopterin (Phel), were purchased from ChromaDex (ChromaDex^®^, Irvine, CA, USA).

### 4.4. Cell Preparation

Cells were seeded on 12-well plates (Sarstedt, Wr. Neudorf, Austria) at an initial density of 1 × 10^6^ cells/mL. After 24 h, the cell suspension was stimulated with coumarin-derivatives at IC50 concentrations. Another group of cells were stimulated with coumarin-derivatives at IC50 concentration with mitoxantrone (cytostatic (+C)) at a concentration of 0.02 µM. We used two controls—cell cultures without stimulators and cell cultures with mitoxantrone at a concentration of 0.02 µM. After 24 h, the cell suspension (from each well) was centrifuged at 800 rpm for 5 min, and the supernatant was discarded.

### 4.5. Quantification of Apoptosis by Annexin V and PI Double Staining

Cells, after 24 h cultivation in culture plates, were exposed to the tested coumarin compounds at the IC50 dose. Then, after 24 h of exposure, cells were centrifuged (800 rpm, time 10 min), the supernatant removed, and the resulting cell pellet was rinsed with 1 mL PBS (WSiS Lublin, Poland), centrifuged again (800 rpm, 5 min) after which the supernatant was removed. 100 μL of the reaction mixture with 20 μL annexin V, 20 μL propidium iodide and 960 μL HEPES incubation buffer (Roche Diagnostics, Basel, Switzerland) were added to the cells. The prepared sample, according to the manufacturer’s instructions, was incubated at room temperature for 15 min in the dark.

Apoptosis analysis was performed using a FlowSight flow cytometer (Amnis Corporation, Seattle, WA, USA) with real-time image analysis using appropriate filters for dyes used: 488 nm excitation, 480–560 nm emission for annexin V and 488 nm excitation, 595–660 nm emission for propidium iodide.

### 4.6. Quantification of Apoptosis by Caspase 3 (PhiPhiLux-G_1_D_2_)

After 24 h cultivation on culture plates, cells were exposed to the tested coumarin compounds at the IC50 dose. Then, after 24 h of exposure, the cells were centrifuged (800 rpm, time 10 min), the supernatant was removed, the resulting cell pellet was rinsed with 1 mL PBS (WSiS Lublin). The pellet was centrifuged a second time (800 rev/min, 5 min), and supernatant removed. 20 μL of PhiPhiLux-G1D2 OncoImmunin reagent (Gaithersburg, Maryland, MD, USA) containing caspase 3 antibodies was added to the cells. The prepared sample, as recommended by the manufacturer, was incubated at 37° for 60 min after which time 1 mL of Binding Buffer was added, samples centrifuged (800 rpm, 5 min) and the supernatant discarded.

Analysis of apoptosis was carried out using a FlowSight flow cytometer (Amnis Corporation, Seattle, WA, USA) with real-time image analysis using appropriate filters for dyes used: 505 nm excitation, 530 nm emission for caspase 3 and 488 nm excitation, 595–660 nm emission for propidium iodide.

### 4.7. Statistical Analysis

Statistical calculations were made using the STATISTICA 12 package (StatSoft Polska Sp. O.o., Kraków, Poland). The results of all experiments are expressed as the mean and standard deviation (SD) of at least three replicates of the experiment. The statistical significance of differences between groups was assessed by Student’s t test. The Kruskal-Willis test was used to analyze the results obtained. The level of statistical significance was set at *p* < 0.05.

## 5. Conclusions

All furanocoumarin derivatives investigated by us induce apoptotic death in the cells of the tested lines HL60, HL60/MX1 and HL60/MX2. After exposure of the cells of the tested lines to furanocoumarin compounds in the presence of mitoxantrone, a different reaction was observed between the two lines tested with phenotypes of multidrug resistance: HL60/MX1 and HL60/MX2.3. It was observed that in cells of the HL60/MX2 line exposed to test compounds in the presence of mitoxantrone, multidrug resistance is induced, expressing a decrease in the percentage of late apoptotic cells after exposure to xanthotoxol, xanthotoxin, isopimpinellin, bergapten, byakangelicin and phellopterin.

In contrast, in HL60/MX1 cells, the inhibition of multidrug resistance has been discontinued, expressed as an increase in the percentage of late apoptotic cells after exposure to xanthotoxol, heraclenin, isopimpinellin, bergapten, byakangelicin and byakangelicol. After exposure to the furanocoumarin tested, a small percentage of necrotic cells were found in the range of 0.5% to 3.75%.

In the light of the obtained results, we can observe a discrepancy between the results obtained with the use of AnnexinV/IP and the use of PhiPhi Lux reagent, which is the active caspase 3, a marker of apoptosis and propidium iodide. Discrepancies in the obtained results may be due to the fact that Annexin V detects a phosphatidylserine translocation from the inside of the cell membrane to the outside. This process may be present not only in cells dying by apoptosis. However, the confirmation of the ongoing apoptosis process is verified by the presence of active caspase 3. It follows that the cells of the HL60/MX1 line are dying not only by apoptosis. Hence their increased percentage in the AnnexinV test after exposure to xanthotoxol, heraclenin, isopimpinellin, bergapten, byakangelicin and byakangelicol, because the percentage of cells with active caspase 3 is significantly lower. In the HL60/MX1 cell line, xanthotoxin causes a higher percentage of apoptotic cells, while the cellular level remains unchanged after exposure to byakangelicol and phellopterin, which may indicate that these compounds do not induce multidrug resistance.

The results obtained significantly expand knowledge on the anti-cancer effect of furanocoumarins and their effect on the induction of apoptosis in derived tumor cell lines from the human hematopoietic system: HL60, HL60/MX1 and HL60/MX2.

## Figures and Tables

**Figure 1 molecules-24-01824-f001:**
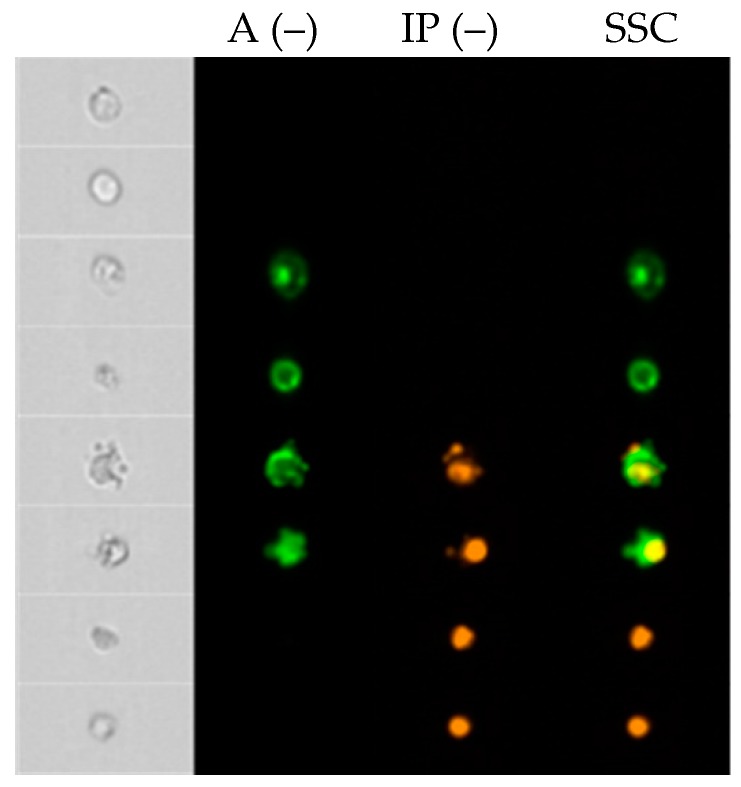
An exemplary cytometric visualization of HL60/MX1 line cells exposed to bergapten in an IC50 dose in the FlowSight instrument in real time.

**Figure 2 molecules-24-01824-f002:**
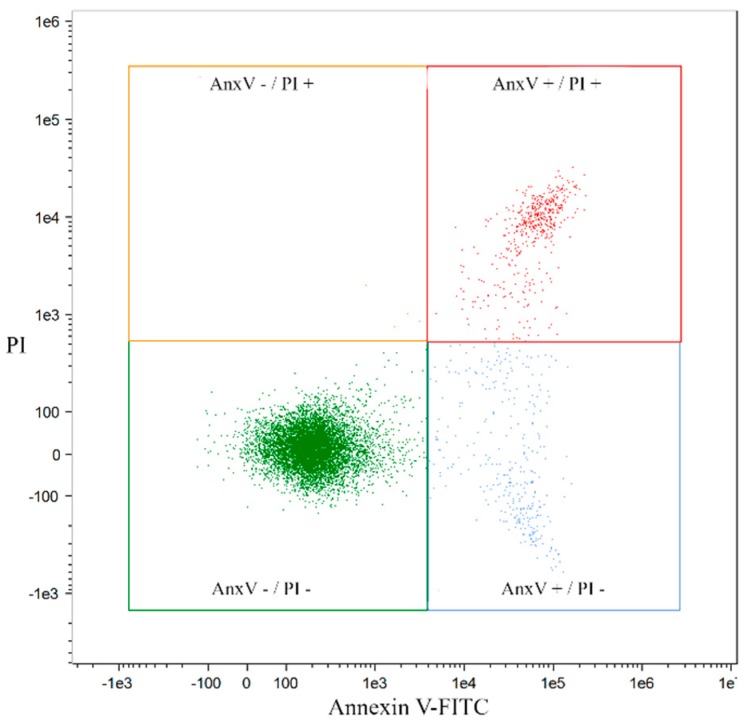
An exemplary “dot plot” of the HL60/MX1 cell line fraction exposed to bergapten at the IC50 dose in the FlowSight instrument in real time.

**Figure 3 molecules-24-01824-f003:**
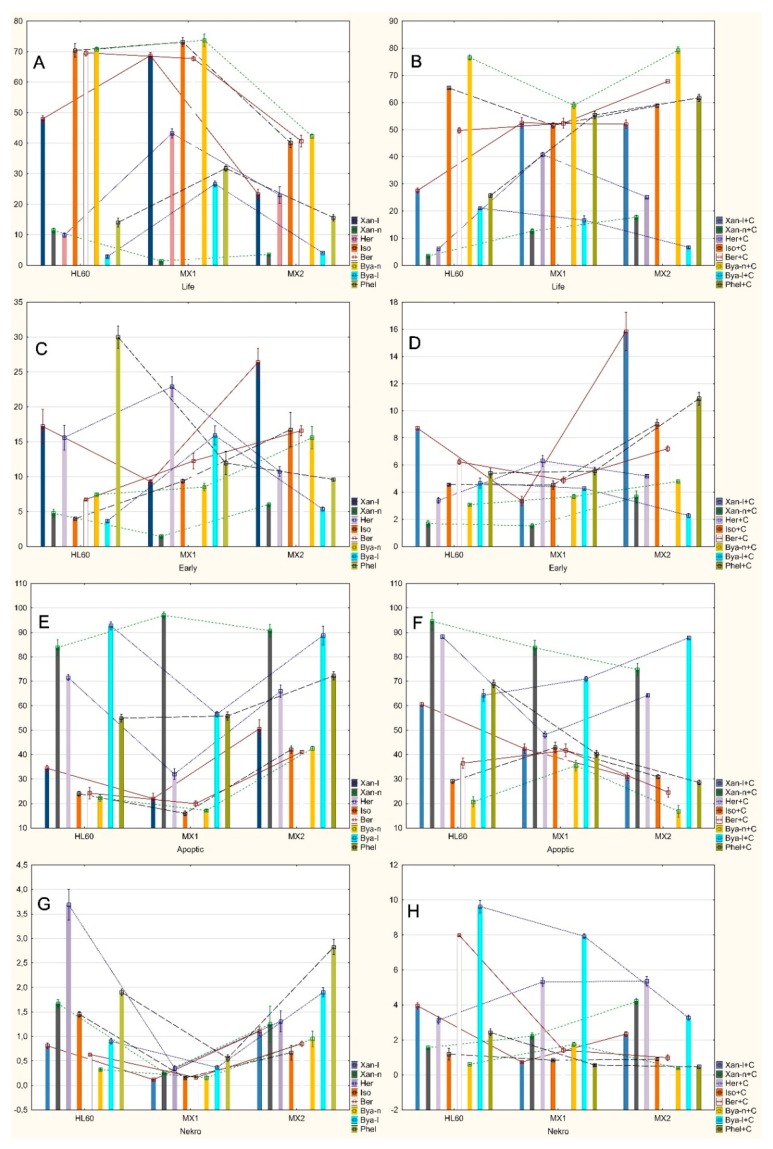
Statistically significant differences in the number of cells between the individual lines: HL60, HL60/MX1 and HL60/MX2 after exposure to the furanocoumarin compounds tested without mitoxantrone (**A**—number of life cells; **C**—number of early apoptotic cells; **E**—number of late apoptotic cells; **G**—number of necrotic cells) or with mitoxantrone (**B**—number of life cells; **D**—number of early apoptotic cells; **F**—number of late apoptotic cells; **H**—number of necrotic cells).

**Figure 4 molecules-24-01824-f004:**
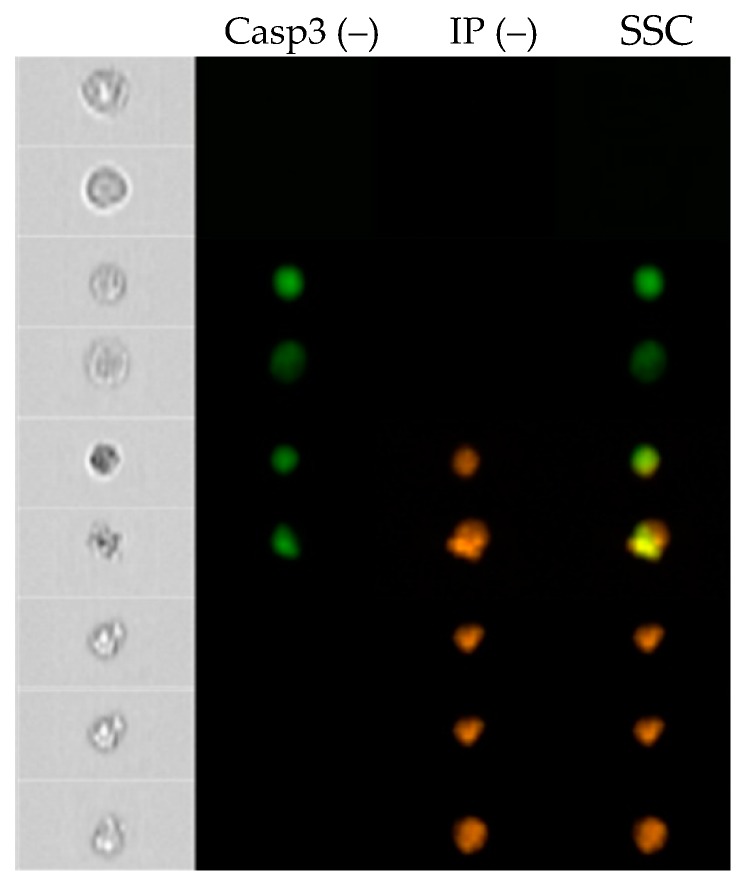
An exemplary cytometric visualization of HL60 cell lines exposed to byakangelicin in an IC50 dose in a real-time FlowSight instrument.

**Figure 5 molecules-24-01824-f005:**
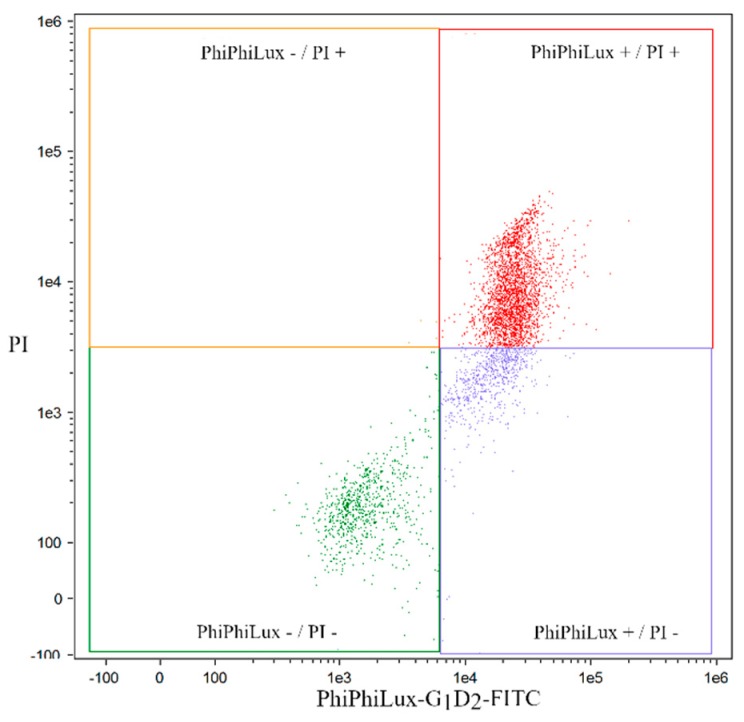
An exemplary “dot plot” illustrating the separation into four fractions of HL60 cell lines exposed to byakangelicin at the IC50 dose.

**Figure 6 molecules-24-01824-f006:**
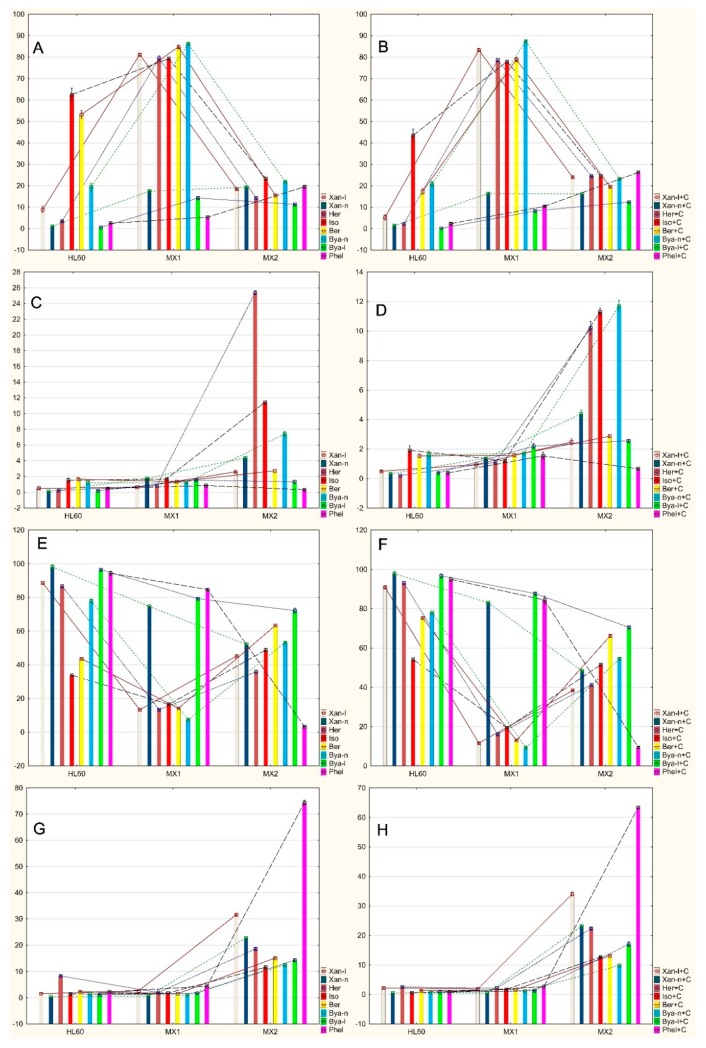
Statistically significant differences in the number of cells between the individual lines: HL60, HL60/MX1 and HL60/MX2 after exposure to the furanocoumarin compounds tested without mitoxantrone (**A**—number of life cells; **C**—number of early apoptotic cells; **E**—number of late apoptotic cells; **G**—number of necrotic cells) or with mitoxantrone (**B**—number of life cells; **D**—number of early apoptotic cells; **F**—number of late apoptotic cells; **H**—number of necrotic cells).

**Table 1 molecules-24-01824-t001:** Statistically significant differences between the lines exposed to the tested furanocoumarin without the presence of mitoxantrone. Results of Kruskal–Wallis test: “Z” values for multiple comparisons, “*p*” values for multiple comparisons. The level of statistical significance was set at *p* < 0.05.

		LIFE			EARLY APOPTOTIC			LATE APOPTOTIC			NECROTIC		
		HL60/ MX1	HL60/ MX2	MX1/ MX2	HL60/ MX1	HL60/ MX2	MX1/ MX2	HL60/ MX1	HL60/ MX2	MX1/ MX2	HL60/ MX1	HL60/ MX2	MX1/ MX2
**Xan-l**	Z	1.341641	1.341641	**2.683282**	1.341641	1.341641	**2.683282**	1.341641	1.341641	**2.683282**	1.341641	1.341641	**2.683282**
	p	0.539137	0.539137	**0.021871**	0.539137	0.539137	**0.021871**	0.539137	0.539137	**0.021871**	0.539137	0.539137	**0.021871**
**Xan-n**	Z	**2.683282**	1.341641	1.341641	1.341641	1.341641	**2.683282**	**2.683282**	1.341641	1.341641	2.385139	0.745356	1.639783
	P	**0.021871**	0.539137	0.539137	0.539137	0.539137	**0.021871**	**0.021871**	0.539137	0.539137	0.051218	1.000000	0.303151
**Her**	Z	**2.683282**	1.341641	1.341641	1.341641	1.341641	**2.683282**	**2.683282**	1.341641	1.341641	**2.683282**	1.341641	1.341641
	p	**0.021871**	0.539137	0.539137	0.539137	0.539137	**0.021871**	**0.021871**	0.539137	0.539137	**0.021871**	0.539137	0.539137
**Iso**	Z	1.043498	1.490712	**2.534210**	1.341641	**2.683282**	1.341641	1.341641	1.341641	**2.683282**	**2.683282**	1.341641	1.341641
	p	0.890153	0.408111	**0.033810**	0.539137	**0.021871**	0.539137	0.539137	0.539137	**0.021871**	**0.021871**	0.539137	0.539137
**Ber**	Z	1.341641	**2.683282**	1.341641	1.341641	**2.683282**	1.341641	1.341641	1.341641	**2.683282**	1.341641	1.341641	**2.683282**
	p	0.539137	**0.021871**	0.539137	0.539137	**0.021871**	0.539137	0.539137	0.539137	**0.021871**	0.539137	0.539137	**0.021871**
**Bya-n**	Z	1.341641	1.341641	**2.683282**	1.341641	**2.683282**	1.341641	1.341641	1.341641	**2.683282**	1.341641	1.341641	**2.683282**
	p	0.539137	0.539137	**0.021871**	0.539137	**0.021871**	0.539137	0.539137	0.539137	**0.021871**	0.539137	0.539137	**0.021871**
**Bya-l**	Z	**2.683282**	1.341641	1.341641	**2.683282**	1.341641	1.341641	2.385139	0.745356	1.639783	1.341641	1.341641	**2.683282**
	p	**0.021871**	0.539137	0.539137	**0.021871**	0.539137	0.539137	0.051218	1.000000	0.303151	0.539137	0.539137	**0.021871**
**Phel**	Z	2.385139	0.745356	1.639783	1.341641	**2.683282**	1.341641	0.447214	2.236068	1.788854	1.341641	1.341641	**2.683282**
	p	0.051218	1.000000	0.303151	0.539137	**0.021871**	0.539137	1.000000	0.076042	0.220915	0.539137	0.539137	**0.021871**

**Table 2 molecules-24-01824-t002:** Statistically significant differences between the lines exposed to the tested furanocoumarin with the presence of mitoxantrone (+C). Results of Kruskal–Wallis test: “Z” values for multiple comparisons, “*p*” values for multiple comparisons. The level of significance was set at *p* < 0.05.

		LIFE			EARLY APOPTOTIC			LATE APOPTOTIC			NECROTIC		
		HL60/ MX1	HL60/ MX2	MX1/ MX2	HL60/ MX1	HL60/ MX2	MX1/ MX2	HL60/ MX1	HL60/ MX2	MX1/ MX2	HL60/ MX1	HL60/ MX2	MX1/ MX2
**Xan-l+C**	Z	2.236068	1.788854	0.447214	1.341641	1.341641	**2.683282**	1.341641	**2.683282**	1.341641	**2.683282**	1.341641	1.341641
	p	0.076042	0.220915	1.000000	0.539137	0.539137	**0.021871**	0.539137	**0.021871**	0.539137	**0.021871**	0.539137	0.539137
**Xan-n+C**	Z	1.341641	**2.683282**	1.341641	0.447214	1.788854	2.236068	1.341641	**2.683282**	1.341641	1.341641	**2.683282**	1.341641
	P	0.539137	**0.021871**	0.539137	1.000000	0.220915	0.076042	0.539137	**0.021871**	0.539137	0.539137	**0.021871**	0.539137
**Her+C**	Z	**2.683282**	1.341641	1.341641	**2.683282**	1.341641	1.341641	**2.683282**	1.341641	1.341641	1.788854	2.236068	0.447214
	p	**0.021871**	0.539137	0.539137	**0.021871**	0.539137	0.539137	**0.021871**	0.539137	0.539137	0.220915	0.076042	1.000000
**Iso+C**	Z	**2.683282**	1.341641	1.341641	0.149071	1.937926	2.086997	**2.683282**	1.341641	1.341641	1.490712	0.745356	0.745356
	p	**0.021871**	0.539137	0.539137	1.000000	0.157897	0.110665	**0.021871**	0.539137	0.539137	0.408111	1.000000	1.000000
**Ber+C**	Z	1.341641	**2.683282**	1.341641	1.341641	1.341641	**2.683282**	1.341641	1.341641	**2.683282**	1.341641	**2.683282**	1.341641
	p	0.539137	**0.021871**	0.539137	0.539137	0.539137	**0.021871**	0.539137	0.539137	**0.021871**	0.539137	**0.021871**	0.539137
**Bya-n+C**	Z	1.341641	1.341641	**2.683282**	1.341641	**2.683282**	1.341641	1.490712	1.043498	**2.534210**	1.341641	1.341641	**2.683282**
	p	0.539137	0.539137	**0.021871**	0.539137	**0.021871**	0.539137	0.408111	0.890153	**0.033810**	0.539137	0.539137	**0.021871**
**Bya-l+C**	Z	1.341641	**2.683282**	1.341641	0.745356	2.385139	1.639783	1.341641	**2.683282**	1.341641	1.341641	**2.683282**	1.341641
	p	0.539137	**0.021871**	0.539137	1.000000	0.051218	0.303151	0.539137	**0.021871**	0.539137	0.539137	**0.021871**	0.539137
**Phel+C**	Z	1.341641	**2.683282**	1.341641	0.447214	2.236068	1.788854	1.341641	**2.683282**	1.341641	1.639783	2.385139	0.745356
	p	0.539137	**0.021871**	0.539137	1.000000	0.076042	0.220915	0.539137	**0.021871**	0.539137	0.303151	0.051218	1.000000

**Table 3 molecules-24-01824-t003:** Statistically significant differences between the lines exposed to the tested furanocoumarin without the presence of mitoxantrone. Results of Kruskal–Wallis test: “Z” values for multiple comparisons, “*p*” values for multiple comparisons. The level of significance was set at *p* < 0.05.

		LIFE			EARLY APOPTOTIC			LATE APOPTOTIC			NECROTIC		
		HL60/ MX1	HL60/ MX2	MX1/ MX2	HL60/ MX1	HL60/ MX2	MX1/ MX2	HL60/ MX1	HL60/ MX2	MX1/ MX2	HL60/ MX1	HL60/ MX2	MX1/ MX2
**Xan-l**	Z	**2.68328**	1.34164	1.341641	1.341641	**2.683282**	1.341641	**2.683282**	1.341641	1.341641	1.341641	**2.683282**	1.341641
	p	**0.02187**	0.53913	0.539137	0.539137	**0.021871**	0.539137	**0.021871**	0.539137	0.539137	0.539137	**0.021871**	0.539137
**Xan-n**	Z	1.34164	**2.68328**	1.341641	1.341641	**2.683282**	1.341641	1.341641	**2.683282**	1.341641	1.341641	**2.683282**	1.341641
	P	0.53913	**0.02187**	0.539137	0.539137	**0.021871**	0.539137	0.539137	**0.021871**	0.539137	0.539137	**0.021871**	0.539137
**Her**	Z	**2.68328**	1.34164	1.341641	1.341641	**2.683282**	1.341641	**2.683282**	1.341641	1.341641	1.341641	1.341641	**2.683282**
	p	**0.02187**	0.53913	0.539137	0.539137	**0.021871**	0.539137	**0.021871**	0.539137	0.539137	0.539137	0.539137	**0.021871**
**Iso**	Z	1.34164	1.34164	**2.683282**	0.447214	2.236068	1.788854	1.341641	1.341641	**2.683282**	1.341641	**2.683282**	1.341641
	p	0.53913	0.53913	**0.021871**	1.000000	0.076042	0.220915	0.539137	0.539137	**0.021871**	0.539137	**0.021871**	0.539137
**Ber**	Z	1.34164	1.34164	**2.683282**	1.341641	1.341641	**2.683282**	1.341641	1.341641	**2.683282**	1.341641	1.341641	**2.683282**
	p	0.53913	0.53913	**0.021871**	0.539137	0.539137	**0.021871**	0.539137	0.539137	**0.021871**	0.539137	0.539137	**0.021871**
**Bya-n**	Z	**2.68328**	1.34164	1.341641	0.447214	2.236068	1.788854	**2.683282**	1.341641	1.341641	1.341641	1.341641	**2.683282**
	p	**0.02187**	0.53913	0.539137	1.000000	0.076042	0.220915	**0.021871**	0.539137	0.539137	0.539137	0.539137	**0.021871**
**Bya-l**	Z	**2.68328**	1.34164	1.341641	2.385139	1.639783	0.745356	1.341641	**2.683282**	1.341641	1.341641	**2.683282**	1.341641
	p	**0.02187**	0.53913	0.539137	0.051218	0.303151	1.000000	0.539137	**0.021871**	0.539137	0.539137	**0.021871**	0.539137
**Phel**	Z	1.34164	**2.68328**	1.341641	1.341641	1.341641	**2.683282**	1.341641	**2.683282**	1.341641	1.341641	**2.683282**	1.341641
	p	0.53913	**0.02187**	0.539137	0.539137	0.539137	**0.021871**	0.539137	**0.021871**	0.539137	0.539137	**0.021871**	0.539137

**Table 4 molecules-24-01824-t004:** Statistically significant differences between the lines exposed to the tested furanocoumarin with the presence of mitoxantrone (+C). Results of Kruskal–Wallis test: “Z” values for multiple comparisons, “*p*” values for multiple comparisons. The level of significance was set at *p* < 0.05.

		LIFE			EARLY APOPTOTIC			LATE APOPTOTIC			NECROTIC		
		HL60/ MX1	HL60/ MX2	MX1/ MX2	HL60/ MX1	HL60/ MX2	MX1/ MX2	HL60/ MX1	HL60/ MX2	MX1/ MX2	HL60/ MX1	HL60/ MX2	MX1/ MX2
**Xan-l+C**	Z	**2.683282**	1.341641	1.341641	1.341641	**2.683282**	1.341641	**2.683282**	1.341641	1.341641	1.341641	1.341641	**2.683282**
	p	**0.021871**	0.539137	0.539137	0.539137	**0.021871**	0.539137	**0.021871**	0.539137	0.539137	0.539137	0.539137	**0.021871**
**Xan-n+C**	Z	2.086997	1.937926	0.149071	1.341641	**2.683282**	1.341641	1.341641	**2.683282**	1.341641	1.341641	1.341641	**2.683282**
	P	0.110665	0.157897	1.000000	0.539137	**0.021871**	0.539137	0.539137	**0.021871**	0.539137	0.539137	0.539137	**0.021871**
**Her+C**	Z	**2.683282**	1.341641	1.341641	1.341641	**2.683282**	1.341641	**2.683282**	1.341641	1.341641	1.341641	1.341641	**2.683282**
	p	**0.021871**	0.539137	0.539137	0.539137	**0.021871**	0.539137	**0.021871**	0.539137	0.539137	0.539137	0.539137	**0.021871**
**Iso+C**	Z	1.341641	1.341641	**2.683282**	1.341641	1.341641	**2.683282**	**2.683282**	1.341641	1.341641	1.341641	**2.683282**	1.341641
	p	0.539137	0.539137	**0.021871**	0.539137	0.539137	**0.021871**	**0.021871**	0.539137	0.539137	0.539137	**0.021871**	0.539137
**Ber+C**	Z	**2.683282**	1.341641	1.341641	0.745356	2.385139	1.639783	**2.683282**	1.341641	1.341641	1.341641	**2.683282**	1.341641
	p	**0.021871**	0.539137	0.539137	1.000000	0.051218	0.303151	**0.021871**	0.539137	0.539137	0.539137	**0.021871**	0.539137
**Bya-n+C**	Z	**2.683282**	1.341641	1.341641	0.447214	2.236068	1.788854	**2.683282**	1.341641	1.341641	1.341641	**2.683282**	1.341641
	p	**0.021871**	0.539137	0.539137	1.000000	0.076042	0.220915	**0.021871**	0.539137	0.539137	0.539137	**0.021871**	0.539137
**Bya-l+C**	Z	1.341641	**2.683282**	1.341641	1.341641	**2.683282**	1.341641	1.341641	**2.683282**	1.341641	1.341641	**2.683282**	1.341641
	p	0.539137	**0.021871**	0.539137	0.539137	**0.021871**	0.539137	0.539137	**0.021871**	0.539137	0.539137	**0.021871**	0.539137
**Phel+C**	Z	1.341641	**2.683282**	1.341641	**2.683282**	1.341641	1.341641	1.341641	**2.683282**	1.341641	1.341641	**2.683282**	1.341641
	p	0.539137	**0.021871**	0.539137	**0.021871**	0.539137	0.539137	0.539137	**0.021871**	0.539137	0.539137	**0.021871**	0.539137

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
