# Peer review of "The Effect of Furanocoumarin Derivatives on Induction of Apoptosis and Multidrug Resistance in Human Leukemic Cells"

_molecules, 2019, doi:10.3390/molecules24091824_

Round 1

Reviewer 1 Report

The manuscript by Kubrak et al examines the effects of furanocourins on apoptosis induction and multidrug resistance in promyelocytic leukemia cell lines: HL60, HL60/MX1, HL60/MX2. The study is carried out with care and the manuscript is generally well written. There is an exceptional amount of data in this study and it can be overwhelming. The authors have used graphs in an interesting way to try and make sense of it all but it still remains difficult at times. Nevertheless the overall study is reasonable and can be of use to those in the field.

Criticisms

1- The authors have previously published a similar type of study examining the expression of multidrug resistant genes in response to coumarins in the same cells (reference #15 in the present manuscript). Even though there is no overlap, the authors have to be very careful here. The authors have used the first two sentences of the abstract of the previous paper almost word for word in the abstract of the present manuscript. Even though it is self-plagiarism, it is not acceptable and the authors should change these sentences and make sure that there are no other similar incidences like this in the remainder of the text.

2- The authors introduce the topic of apoptosis and discuss various aspects of it in the introduction (lines 53 – 58). They include no citation for most of their statements, so they should include a general reference such as (Portt et al 2011. Anti-apoptosis and cell survival: a review, Biochim Biophys Acta 1813: 238-259).

3- The authors discuss the usefulness of different coumarin derivatives (lines 398 – 403). They should also include a more recent reference on the topic (Zhu and Jiang 2018. Pharmacological and Nutritional Effects of Natural Coumarins and Their Structure-Activity Relationships. Mol Nutr Food Res. May 11:e1701073. doi: 10.1002/mnfr.201701073).

Author Response

Author's Reply to the Review Report (Reviewer 1)

x) English language and style are fine/minor spell check required 

Manuscript has been read and corrected once again by Native Speaker.

1- The authors have previously published a similar type of study examining the expression of multidrug resistant genes in response to coumarins in the same cells (reference #15 in the present manuscript). Even though there is no overlap, the authors have to be very careful here. The authors have used the first two sentences of the abstract of the previous paper almost word for word in the abstract of the present manuscript. Even though it is self-plagiarism, it is not acceptable and the authors should change these sentences and make sure that there are no other similar incidences like this in the remainder of the text.

Answer to Reviewer1:

- Yes, there really were repeats in three places. Manscript has been checked and improved.

2- The authors introduce the topic of apoptosis and discuss various aspects of it in the introduction (lines 53 – 58). They include no citation for most of their statements, so they should include a general reference such as (Portt et al 2011. Anti-apoptosis and cell survival: a review, Biochim Biophys Acta 1813: 238-259).

Answer to Reviewer1:

- Yes, indeed – thank you for the citation- It has been added.

3- The authors discuss the usefulness of different coumarin derivatives (lines 398 – 403). They should also include a more recent reference on the topic (Zhu and Jiang 2018. Pharmacological and Nutritional Effects of Natural Coumarins and Their Structure-Activity Relationships. Mol Nutr Food Res. May 11:e1701073. doi: 10.1002/mnfr.201701073).

Answer to Reviewer1:

Yes, indeed – thank you for the  citation- It has been added.

Reviewer 2 Report

Q1. Did the furanocoumarins have the same mechanism as the chemical drugs?

Q2. Did the normal cells go through apoptosis under these compounds? If so, how serious?

Q3. Is there any way the authors can determine the drug molecules bind which protein target?

Author Response

Author's Reply to the Review Report (Reviewer 2)

(x)           Does the introduction provide sufficient background and include all relevant references?

Answer to Reviewer2:

- The introduction has been improved.

Q1. Did the furanocoumarins have the same mechanism as the chemical drugs?

Answer to Reviewer2:

Furanocoumarins, due to their chemical structure (furan ring and piron ring) have a specific way of interacting, they form monoadducts with a DNA strand. The effect of these changes is disruption of DNA transcription, disruption of genetic information transmission, changes in gene and protein expression, and in the final stage inhibition of cellular hyperproliferation [eg. Spielmann HP, Dwyer TJ, Sastry SS, Hearst JE, Wemmer DE. DNA structural reorganization upon conversion of a psoralen furan-side monoadduct to an interstrand cross-link: implications for DNA repair. Proc Natl Acad Sci U S A. 1995 Mar 14;92(6):2345-9; Couvé-Privat, S., Macé, G.;Rosselli, F.;Saparbaev, M. , Psoralen-induced DNA adducts are substrates for the base excision repair pathway in human cellsNucleic Acids Res. 2007 Sep; 35(17): 5672–5682].

Several furanocoumarins have also been tested for their ability to inhibit human P450 activity [Luke, L. Koenigs, William F. Trager. Mechanism-Based Inactivation of P450 2A6 by Furanocoumarins. Biochemistry, 1998, 37 (28), pp 10047–10061.DOI: 10.1021/bi980003c].

Q2. Did the normal cells go through apoptosis under these compounds? If so, how serious?

Answer to Reviewer2:

Of course, furanocoumarins induce apoptosis of normal cells, more or less cytotoxic.

In our study, [Bogucka-Kocka A. Evaluation of the interaction of selected plant-related proteins on the expression of genes regulating the process of apoptosis of human hematopoietic cell cancer cells, 2010, Medical University of Lublin, Habilitation dissertation, pages322, ISBN 978-83-62606-03-0) the IC50 value of selected furanocoumarins for many tumor lines (1301, U266, ML1, EOL, J45) as well as for the normal WILCL line was determined. The xanthotoxin and xanthotoxol were active cytotoxically (the highest IC50 value).

Toxicity in vivo is so permissible that it is used in the treatment of various diseases, e.g. skin diseases (vitiligo, psoriasis), cutaneous lymphomas, eczema.

Research is conducted on the usefulness of furanocoumarins in the treatment of neurodegenerative diseases and cancer diseases, e.g. leukemia.

Q3. Is there any way the authors can determine the drug molecules bind which protein target?

Answer to Reviewer2:

In our work [Kubrak, T.; Bogucka-Kocka, A.; Komsta, Ł. et. al. Modulation of multidrug resistance gene expression by coumarin derivatives in human leukemic cells, Oxid. Med. Cell Longev. 2017, Article ID 5647281], we showed that furanocoumarins reduce the expression of the BCRP2, MRP and LRP genes and increase the expression of MDR1, which in turn translates into the synthesis of target proteins encoded by these genes.

We have performed studies on how furanocoumarins affect the expression of the target protein. The results obtained are the subject of a future publication.
